

# A hyaena on stilts: comparison of the limb morphology of *Ictitherium ebu* (Mammalia: Hyaenidae) from the Late Miocene of Lothagam, Turkana Basin, Kenya with extant Canidae and Hyaenidae

Julien van der Hoek[1] and Lars Werdelin[2]

[1] Department of Earth and Environmental Sciences, University of Manchester, Manchester, Greater Manchester, United Kingdom
[2] Department of Palaeobiology, Swedish Museum of Natural History, Stockholm, Stockholm County, Sweden

Corresponding author
Julien van der Hoek,
julien.vanderhoek@manchester.ac.uk

## ABSTRACT

The long, gracile morphology of the limb bones of the Late Miocene hyaenid *Ictitherium ebu* has led to the hypothesis that this animal was cursorial. The forelimb and femur of the holotype were compared with specimens of extant Hyaenidae and Canidae. Two morphometric methods were used. The first used measurements to calculate indices of different morphological characters. The second method involved capturing photographs of the anterior distal humerus of each specimen, mapping six landmarks on them, and calculating truss distances. These distances represent a schematic reproduction of the elbow. Multivariate statistical analysis primarily separated the data based on taxonomy, yet locomotor and habitat categories were also considered. *Ictitherium ebu* has an overall morphology similar to that of the maned wolf and a distal humerus reminiscent of that of the aardwolf. The long, gracile limb bones of *I. ebu* are suggested to be adaptations for pouncing on prey, for locomotor efficiency, and for looking over the tall grass of the open environments the animal lived in, much like the present-day maned wolf.

## INTRODUCTION

Hyaenidae is a family of considerable palaeontological interest, due to their occurrence in many Miocene-Pleistocene sites in Eurasia, Africa and North America (*Kurtén, 1968*; *Werdelin & Solounias, 1991*; *Werdelin & Solounias, 1996*; *Turner, Antón & Werdelin, 2008*) and the importance of the three larger species of hyaenids for their respective ecosystems (*Rieger, 1981*; *Mills, 1982*; *Turner, Antón & Werdelin, 2008*; *Hayssen & Noonan, 2021*). The pattern of their evolution in Eurasia is relatively clear. They started off as viverrid- and herpestid-like forms, which evolved into canid-like forms, then branched off into cursorial forms, as well as the bone crushing animals we know today (*Werdelin &*

*Solounias, 1991*; *Werdelin & Solounias, 1996*; *Turner, Antón & Werdelin, 2008*). During the Miocene-Pliocene mammalian turnover the number of cursorial, canid-like hyaenid species decreased, while the number of Canidae increased. Few bone-crushing hyaenids are known from the latest Miocene, whereas they show up more prominently during the Pliocene. During the Pleistocene Hyaenidae became increasingly adapted to bone-crushing, with the more cursorial morphotypes disappearing.

This pattern is contrasted with community patterns in Late Miocene Africa, which are not as well understood, especially when compared to the evolutionary pattern of Eurasia (*Werdelin, 2003*; *Werdelin & Peigné, 2010*). The carnivore material from Lothagam (Kenya) may allow for such an investigation to take place (*Werdelin, 2003*), in part through better understanding of the ecological roles of the species found in this material.

Lothagam is a Miocene-Pliocene site located near Lake Turkana in Turkana County, Kenya (Fig. 1). It has been dated from 8 to slightly less than 4 Ma (*Leakey, 2003*). The exceptional preservation of fossils at the site is due to the initial accumulation of sediment from a large meandering river system. Massive faulting led to the formation of a horst, which has kept the fossils from being buried. The resistance of the fine-grained matrix of most of the site has also contributed to the preservation. It is an important site for mammal palaeontology, as it is the type site for seven genera and 21 species of mammal.

The carnivoran fauna of Lothagam includes Amphicyonidae, Mustelidae, Viverridae, Hyaenidae, Felidae and Canidae and resembles Langebaanweg in South Africa in overall structure (*Werdelin, 2003*). The hyaenid fauna of Lothagam includes *Ictitherium ebu*, *Hyaenictitherium cf. H. parvum*, *cf. Hyaenictis sp.*, and *Ikelohyaena cf. I. abronia*. The first two species have been identified as jackal/wolf-like ecomorphotypes, *Hyaenictis* as a genus of cursorial meat eaters, and *I. abronia* as a transitional bone cracker (*Turner, Antón & Werdelin, 2008*).

The holotype of *I. ebu*, KNM-LT 23145, was found in the Lower Nawata Member of the Nawata Formation (*Werdelin, 2003*). This formation represents fluvial facies that shows fluctuations in water balance and subsidence rate (*Feibel, 2003*). The Lower Nawata Member is characterised by conglomerate beds of varying thickness, sandstones, and mudstones, together with volcanic detritus with a large amount of intercalated altered distal tephra. The age of the Lower Nawata is 7.4 ± 0.1 to 6.5 ± 0.1 Ma (*McDougall & Feibel, 2003*). The palaeosols of the Lower Nawata mainly represent relatively open grassland, gallery woodland and thornbush savanna (*Wynn, 2003*). Pure grassland has not been recorded in the palaeosols, meaning that it was likely not long-lived if present. The Lower Nawata is characterised by the presence of Bovidae, Hippopotamidae, Suidae and Cercopithecidae, which indicate a well-vegetated habitat (*Leakey & Harris, 2003*).

Of the four hyaenids found at Lothagam, *I. ebu* is by far the best preserved, as it includes both postcranial and craniodental material (*Werdelin, 2003*). It has a dentition that is seemingly adapted for a more hypercarnivorous lifestyle than other members of *Ictitherium*, but is more hypocarnivorous compared to some other Ictitheres. This lifestyle seems to be supported by the notably long and slender limbs of the species, which could be interpreted as an adaptation for cursoriality. The ecology and behaviour of canid-like hyaenids has been mentioned as needing further investigation (*Turner, Antón & Werdelin,*

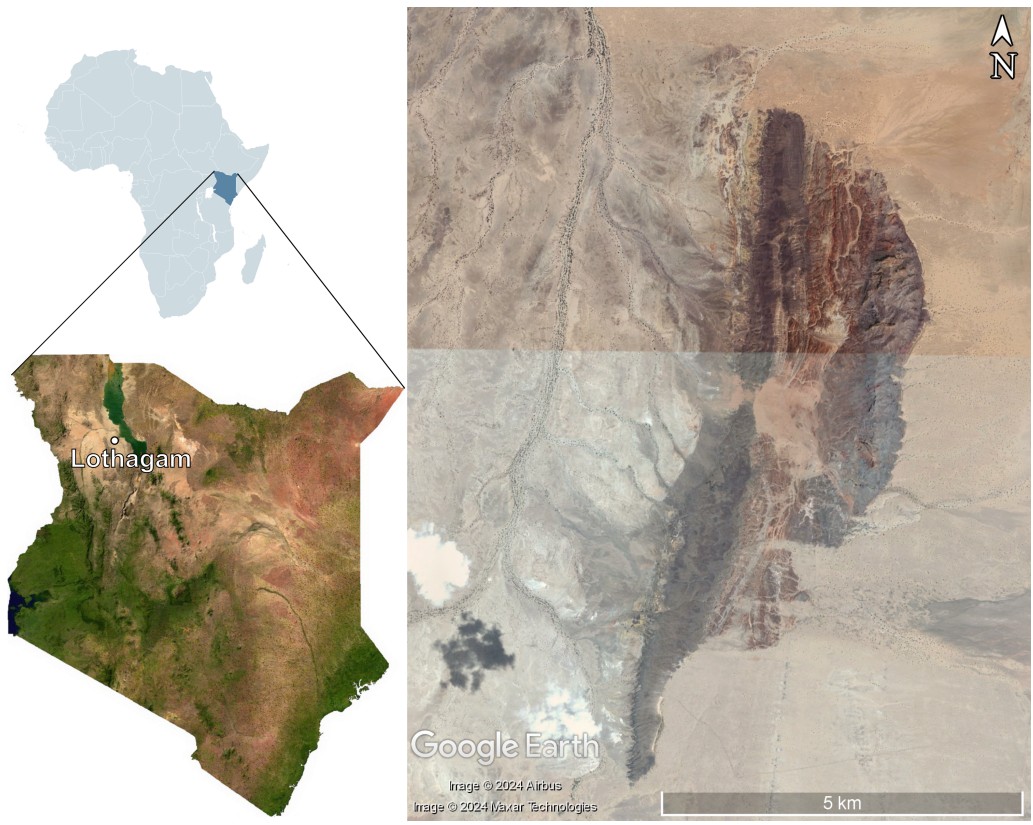

**Figure 1** **Map of Kenya showing the location of Lothagam, with the fossil site on the right.** Maps of Africa and Kenya were obtained from *Wikimedia Commons (2009)* and *Wikimedia Commons (2023)*. The map of Lothagam was obtained from Google Earth Pro (Maxar Technologies and Airbus).

*2008*).The present study provides insight into the ecology of a canid-like hyaenid, as well as into the apparent cursorial adaptations of *I. ebu*.

Extant carnivorans can be classified into different locomotor categories, such as arboreal, scansorial, terrestrial and semi-fossorial, as shown by *Van Valkenburgh (1985)* and *Van Valkenburgh (1987)*. By comparing body mass and skeletal measurements using bivariate and multivariate analysis, it was found that skeletal indicators can predict locomotor behaviour in extant carnivorans. This technique was also applied to extinct carnivorans, with partial success. *Van Valkenburgh (1987)* noted that the characters that define locomotor behaviour in extant carnivorans might differ from those of extinct carnivorans. However, if the biomechanical function of each part of an extinct carnivoran is understood, then it should be possible to reconstruct its locomotor behaviour as well.

*Samuels, Meachen & Sakai (2013)* and *Andersson (2004)* expanded upon the methods of *Van Valkenburgh (1985)* and *Van Valkenburgh (1987)*. *Samuels, Meachen & Sakai (2013)* expanded on the skeletal indicators and added cursorial and semi-aquatic categories. *Andersson (2004)* applied truss analysis (*Strauss & Bookstein, 1982*) to the distal humerus to separate grappling from non-grappling predators. These two methods are here combined to test the hypothesis that *I. ebu* was adapted for cursoriality.

Determining the ecomorphology of *I. ebu* will not only shed light on the ecological role of this species but can ultimately contribute to a better understanding of the ecology of the Late Miocene communities of Lothagam and eastern Africa as a whole. Furthermore, the comparisons might reveal whether *I. ebu* had an ecomorphology that converges on the Canidae, among which many species are cursorially adapted (*Samuels, Meachen & Sakai, 2013*). *Ictitherium* was part of the jackal and wolf-like meat eater ecomorph of *Werdelin & Solounias (1996)* (see also *Turner, Antón & Werdelin, 2008*; *Coca-Ortega & Pérez-Claros, 2019*). This ecomorphology would then be in line with hyaenids in Eurasia having a more cursorial, canid-like morphology before being replaced by canids (*Werdelin & Turner, 1996*; *Turner, Antón & Werdelin, 2008*).

The objectives of this study are twofold: (1) to create models capable of predicting the ecomorphology of *I. ebu*. (2) to test the hypothesis that *I. ebu* was cursorial.

## MATERIALS & METHODS

To be able to test if *I. ebu* was adapted for cursoriality, the appendicular skeletons of different extant carnivorans, of which the ecomorphology is known, were compared to the holotype of *I. ebu*, KNM-LT 23145 from the Nawata Formation. The approach of *Samuels, Meachen & Sakai (2013)* was adapted to study the appendicular skeleton as a whole, while the method of *Andersson (2004)* was used to study the distal humerus.

Because *I. ebu* is hypothesized to be a cursorial hyaenid, with cursorial adaptations similar to those of Canidae (*Turner, Antón & Werdelin, 2008*), only canids and hyaenids were selected. The four extant Hyaenidae were included in the study as they represent the closest living relatives to *I. ebu*. While the spotted hyaena (*Crocuta crocuta*), the striped hyaena (*Hyaena hyaena*), and the brown hyaena (*Parahyaena brunnea*) are cursors, the aardwolf (*Proteles cristatus*) has a more generalist locomotor type (*Mills, 1982*; *Spoor & Badoux, 1988*; *Koehler & Richardson, 1990*; *Samuels, Meachen & Sakai, 2013*; *Hayssen & Noonan, 2021*). The Canidae in this study include a wide range of sizes from the small red fox (*Vulpes vulpes*) to the medium sized coyote (*Canis latrans*) and side-striped jackal (*Lupullela adusta*), to the large wolf (*Canis lupus*). The maned wolf (*Chrysocyon brachyurus*) was included for its morphology, because its long, slender limbs bear a resemblance to those of *I. ebu*.

### Specimen collection

The remains used for the study of *Ictitherium ebu* include the manus, radius, ulna, humerus and femur of a cast of specimen KNM-LT 23145 from the Late Miocene of Lothagam, Kenya, housed in the National Museums of Kenya (NMK) (Fig. 2). A tibia is present as well, but it is broken at the diaphysis, which limits its relevance to this study. The collections of Naturhistoriska Riksmuseet (NRM), Museum für Naturkunde (ZMB), Senckenberg Naturmuseum (SMF), Alexander Koenig Zoological Research Museum (ZFMK), Royal Museum for Central Africa (RMCA), Naturalis Biodiversity Center (RMNH) and La Specola (MZUF) were visited to collect photographs and measurements of 79 specimens of extant species for comparison with KNM-LT 23145 (Table 1). Adult specimens of both sexes were chosen, with a preference for wild-caught individuals.

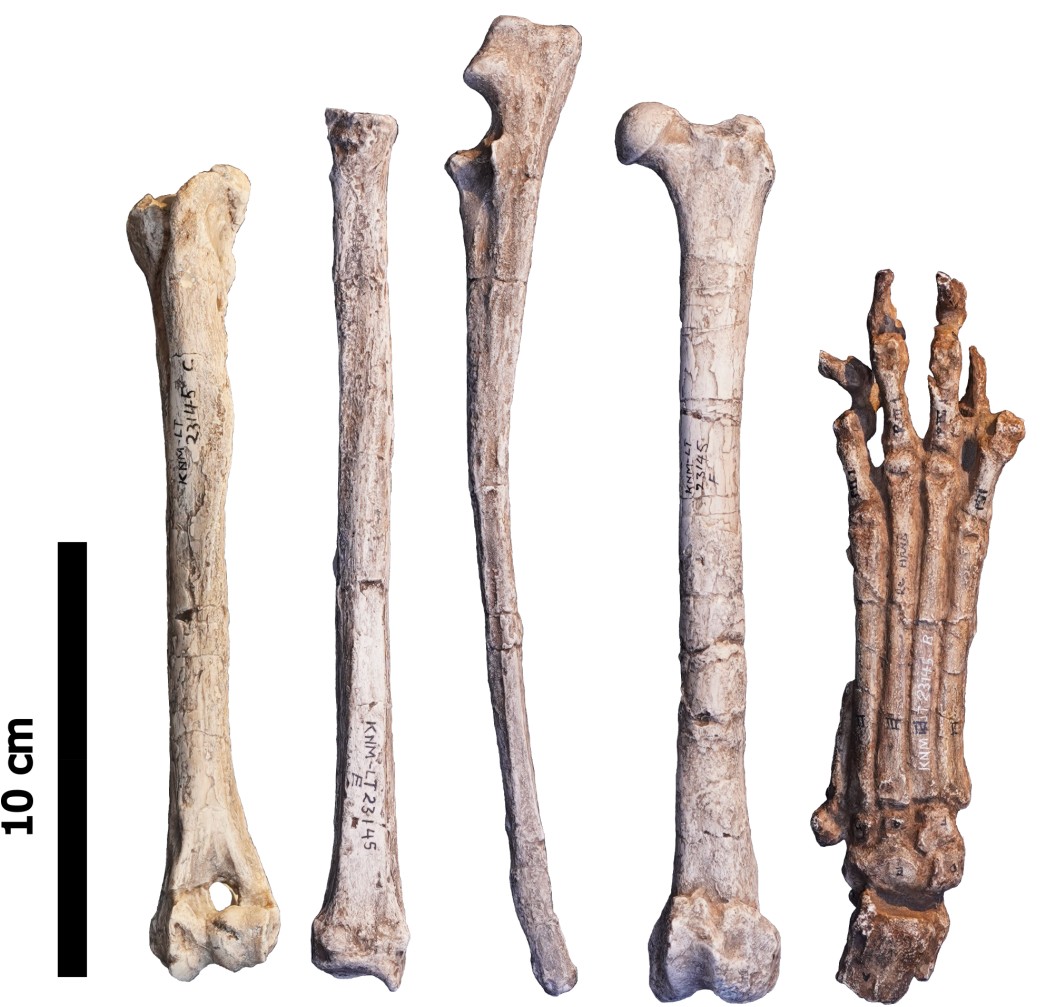

**Figure 2** The left anterior humerus, left anterior radius, left lateral ulna, left anterior femur and dorsal right manus of specimen KNM-LT 23145 (*Ictitherium ebu*). Scalebar is 10 cm.

Only three of the nine spotted hyaena specimens used in this study had completely fused epiphyses (ZMB MAM 7784, ZMB MAM 13295 and ZMB MAM 47515), which is normally the main indicator of an adult animal. However, according to *Egeland, Egeland & Bunn (2008)*, sub-adulthood is characterised by unfused or partially fused epiphyses with solid bone surfaces. Sponginess only occurs near-epiphyses. Adults have epiphyses that are mostly or completely fused, with the entire surface of the bone being solid. Specimens ZMB MAM 14818, ZMB MAM 16575, and ZMB MAM 82415 have fused epiphyses with entirely solid bone surfaces; only thin grooves show the epiphyses to not be entirely fused. They could thus be interpreted as adult specimens. ZMB MAM 82413, ZMB MAM 82471 and ZMB MAM 82516 do not have fused epiphyses, but do not show much spongy bone around the epiphyses. These specimens can then be interpreted as older sub-adults. The striped hyaena ZMB MAM 82363 also had a humerus that was not fully fused, which would indicate a subadult if interpreted in the same way as the spotted hyaena specimens

**Table 1  The species used in this study including number of specimens, their locomotor type, habitat and body mass.**

| Family | Genus | Species | Common name | Locomotor type | Habitat | Body mass (kg) | n |
|---|---|---|---|---|---|---|---|
| Hyaenidae | *Ictitherium*[†] | *ebu*[†] | n.a. | Cursorial[*] | Mixed[**] | 10–15[***] | 1 |
| Hyaenidae | *Proteles* | *cristatus* | Aardwolf | Generalist[1,2] | Open[2] | 47–79[2] | 10 |
| Hyaenidae | *Hyaena* | *hyaena* | Striped hyaena | Cursorial[1,3] | Open[4] | 22–55[4] | 12 |
| Hyaenidae | *Parahyaena* | *brunnea* | Brown hyaena | Cursorial[1,5] | Open[5] | 28–47.5[5] | 2 |
| Hyaenidae | *Crocuta* | *crocuta* | Spotted hyaena | Cursorial[1,6] | Mixed[7] | 47–79[6] | 9 |
| Canidae | *Lupulella* | *adusta* | Side-striped jackal | Cursorial[1,8] | Mixed[8] | 8–10[14] | 10 |
| Canidae | *Vulpes* | *vulpes* | Red fox | Cursorial[1,8] | Mixed[8] | 3–14[15] | 10 |
| Canidae | *Canis* | *lupus* | Wolf | Cursorial[1,9] | Mixed[9] | 18–80[9] | 12 |
| Canidae | *Chrysocyon* | *brachyurus* | Maned wolf | Generalist[1,10,11] | Open[12] | 23[16] | 12 |
| Canidae | *Canis* | *latrans* | Coyote | Cursorial[1,13] | Mixed[13] | 7–20[13] | 1 |

Notes.

[†] Extinct.

[*] *I ebu* was hypothesised to be cursorial by *Werdelin (2003)*.

[**] The Lower Nawata where *I. ebu* was found represents a mixed habitat (*Wynn 2003*).

[***] Body mass estimated by *Werdelin (2003)*.

1. *Samuels, Meachen & Sakai (2013)*. 2. *Koehler & Richardson (1990)*. 3. *Spoor & Badoux (1988)*. 4. *Rieger (1981)*. 5. *Mills (1982)*. 6. *Hayssen & Noonan (2021)*. 7. *Matthews (1939)*. 8. *Sillero-Zubiri, Hoffmann & Macdonald (2004)*. 9. *Mech (1974)*. 10. *Hildebrand (1954)*. 11. *Janis & Wilhelm (1993)*. 12. *Coelho et al. (2018)*. 13. *Bekoff (1977)*. 14. *Bingham & Purchase (2002)*. 15. *Nowak & Paradiso (1983)*. 16. *Dietz (1985)*.

(*Egeland, Egeland & Bunn, 2008*). However, it does not have any measurements that are below the range of the other specimens. Only the radial diameter is smaller than the others, but three specimens have equally small radial diameters. Truss distances all fall within the range of other specimens.

MZUF 13354 (wolf) illustrates the difference between zoo animals and wild animals quite accurately. In the species dataset, MZUF 13354 has the most extreme value for every single index, with eight outliers. The short bones of this animal are the likely cause of these values.

## Linear morphometrics

Specimens were measured according to the protocol of *Samuels & Van Valkenburgh (2008)* and *Samuels, Meachen & Sakai (2013)*, with an added measurement for the midshaft mediolateral diameter of the radius (RD) (Fig. 3, Table 2). The measurement for FGT was carried out differently from the one in *Samuels & Van Valkenburgh (2008)* and *Samuels, Meachen & Sakai (2013)*, where the height of the greater trochanter of the femur was measured vertically instead of diagonally.

The measurements were taken with vernier callipers for measurements up to 15 cm. Measuring tape was used for measurements above 15 cm in all museums except for the Museum für Naturkunde in Berlin, where the measurements were carried out with larger callipers. Measurements were recorded to the nearest 0.1 cm (Table S1).

## Index calculations

The measurements were converted into indices (*Samuels & Van Valkenburgh, 2008*; *Samuels, Meachen & Sakai, 2013*) in Excel 16.0.15330.20260 (Table 3, Dataset S1). The manus proportions index was excluded from further analysis due to a lack of measurements. The radial robustness index, metacarpal/radial index, humeral/femoral index and

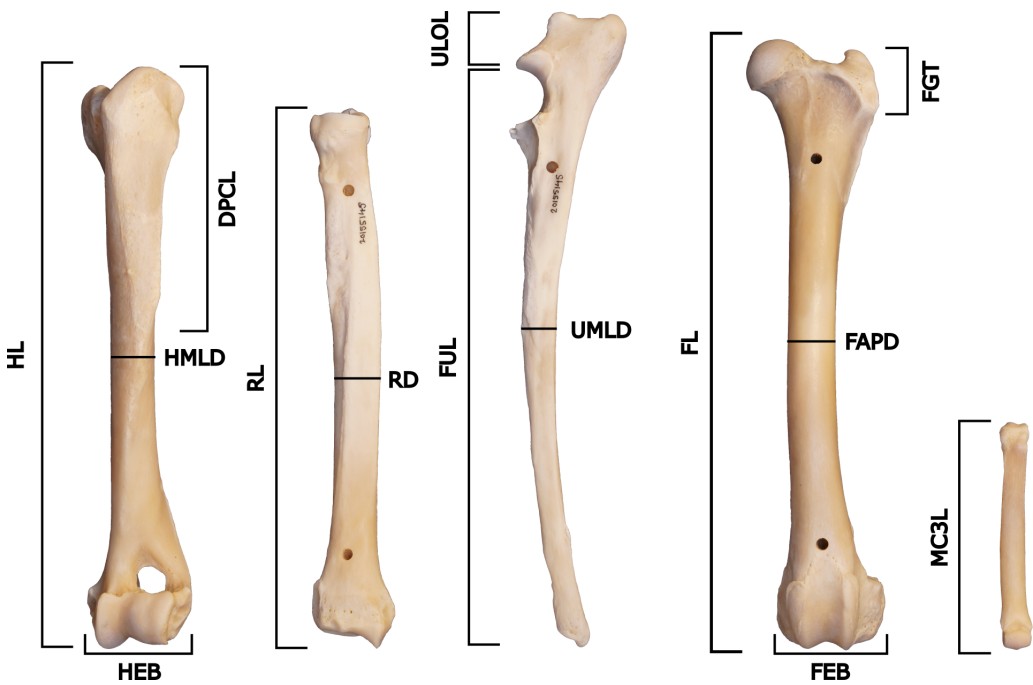

**Figure 3   Measurements of the postcrania used in the project, based on** *Samuels & Van Valkenburgh (2008)* **and** *Samuels, Meachen & Sakai (2013)***.** The specimen figured is NRM 20155145 (*Canis lupus*) and shows the left anterior humerus, left anterior radius, left lateral ulna, left anterior femur and dorsal right manus (not to scale).

**Table 2   Measurements according to** *Samuels & Van Valkenburgh (2008)* **and** *Samuels, Meachen & Sakai (2013)***, with RD added.**

| Measurement | Description |
| --- | --- |
| HL | Greatest length of the humerus |
| HMLD | Midshaft mediolateral diameter of the humerus |
| DPCL | Length of the deltopectoral crest |
| HEB | Epicondylar breadth of the distal humerus |
| RL | Greatest length of the radius |
| RD | Midshaft mediolateral diameter of the radius |
| FUL | Functional length of the ulna |
| UMLD | Midshaft mediolateral diameter of the ulna |
| ULOL | Length of the olecranon process of the ulna |
| MC3L | Greatest length of metacarpal 3 |
| FL | Greatest length of the femur |
| FAPD | Midshaft anteroposterior diameter of the femur |
| FGT | Height of the greater trochanter of the femur |
| FEB | Epicondylar breadth of the distal femur |

metacarpal/humeral index were included so as to take into account the metacarpal III measurements and the relationship between the humerus and the femur.

**Table 3   Indices following *Samuels & Van Valkenburgh (2008)* and *Samuels, Meachen & Sakai (2013)*, with RRI, MCRI, MCHUM and HFI added and MANUS removed.**

| Index | Description |
| --- | --- |
| Shoulder moment index (SMI) | Deltopectoral crest length divided by functional length of the humerus (DPCL/HL). Indicates mechanical advantage of the deltoid and pectoral muscles acting across the shoulder joint. |
| Brachial index (BI) | Functional length of the radius divided by functional length of the humerus (RL/HL). Indicates relative proportions of proximal and distal elements of the forelimb. |
| Humeral robustness index (HRI) | Mediolateral diameter of humerus divided by functional length of the humerus (HMLD/HL). Indicates robustness of the humerus and its ability to resist bending and shearing stresses. |
| Humeral epicondylar index (HEI) | Epicondylar breadth of humerus divided by functional length of the humerus (HEB/HL). Indicates relative area available for the origins of the forearm flexors, pronators, and supinators. |
| Olecranon length index (OLI) | Olecranon process length divided by functional length of the ulna (ULOL/FUL). Indicates relative mechanical advantage of the triceps brachii and dorsoepitrochlearis muscles used in elbow extension. This is identical to the index of fossorial ability used by Hildebrand (1985). |
| Ulnar robustness index (URI) | Mediolateral diameter of ulna divided by functional length of the ulna (UMLD/FUL). Indicates robustness of the ulna and its ability to resist bending and shearing stresses, and relative area available for the origin and insertion of forearm and manus flexors, pronators, and supinators. |
| Femoral robustness index (FRI) | Anteroposterior diameter of femur divided by functional length of the femur (FAPD/FL). Indicates robustness of the femur and its ability to resist bending and shearing stresses (AP diameter is used due to transverse expansion of the femora in semiaquatic rodents). |
| Gluteal index (GI) | Length of distal extension of the greater trochanter of the femur divided by functional length of the femur (FGT/FL). Indicates relative mechanical advantage of the gluteal muscles used in retraction of the femur. |
| Femoral epicondylar index (FEI) | Epicondylar breadth of femur divided by the functional length of the femur (FEB/FL). Indicates relative area available for the origins of the gastrocnemius and soleus muscles used in extension of the knee and plantar-flexion of the pes. |
| Radial robustness index (RRI) | Greatest length of the radius divided by the midshaft mediolateral diameter of the radius (RD/RL). Indicates robustness of the radius and its ability to resist bending and shearing stresses. |
| Metacarpal radial index (MCRI) | Greatest length of metacarpal 3 divided by the functional length of the radius (MC3L/RL). Indicates relative proportions of the third metacarpal compared to the radius. |

**Table 3** (*continued*)

| Index | Description |
|---|---|
| Metacarpal humeral index (MCHUM) | Greatest length of metacarpal 3 divided by the functional length of the humerus (MC3L/HL). Indicates relative proportions of the third metacarpal compared to the length of the humerus. |
| Humeral femoral index (HFI) | Functional length of the radius divided by the functional length of the femur(HL/FL). Indicates relative proportions of the humerus compared to the femur. |

For initial interpretation, preliminary boxplots of the indices plotted against species were created. These boxplots revealed some outliers in the measurements, among which some are measurement errors. These values were often far too extreme to be viewed as simple extremes in the data. For example, the humeral epicondylar breadth of ZMB MAM 82516 was measured to be half that of the other specimens, while the length is within the range of the other specimens. With the use of ImageJ 1.53n to approximate what measurement would have been obtained on-site, it was determined that 10 outliers needed to be removed from the dataset (Table S2).

Aside from these outliers, there were some missing values in the dataset. For these missing values, means were interpolated in Excel 16.0.15330.20260 for species with missing indices by using the mean of the same index for the other specimens of the same species. Boxplots were recreated for the final analysis, with all of these changes incorporated.

## Image collection

The distal humerus of each specimen was photographed in anterior view. The camera was set to have an ISO of 200/250, an aperture of F8–F10, after which the shutter speed was adjusted for brightness. Photographing was carried out with flash. The scale bar was held in place using two alligator clips on 4-way swivels, at the height of the specimen. Some photographs appeared overexposed after data collection. These images were edited using GIMP 2.10.30 with lowering of highlights and lowering the point at which highlights turn to white. The specimens edited in this manner were SMF 97379, SMF 97380 and ZMB MAM 89495. Some photographs were removed from the analysis due to misalignment of the anterior view or deformation of the humerus, see Table S3.

## Truss creation

Using TpsUtil64 1.81 a TPS file was created from the photographs acquired during data collection. The TPS file was imported into TpsDig264 2.32, labelled and scaled according to the scalebar in each image. Six landmarks were assigned to each specimen following *Andersson (2004)* (Fig. 4). The coordinates of the landmarks for each specimen were exported to Excel 16.0.15330.20260 using MorphoJ 1.07a (Table S4), after which the distances of the truss were calculated (Dataset S1). These truss distances combine into a schematic representation of the size and shape of the elbow joint.

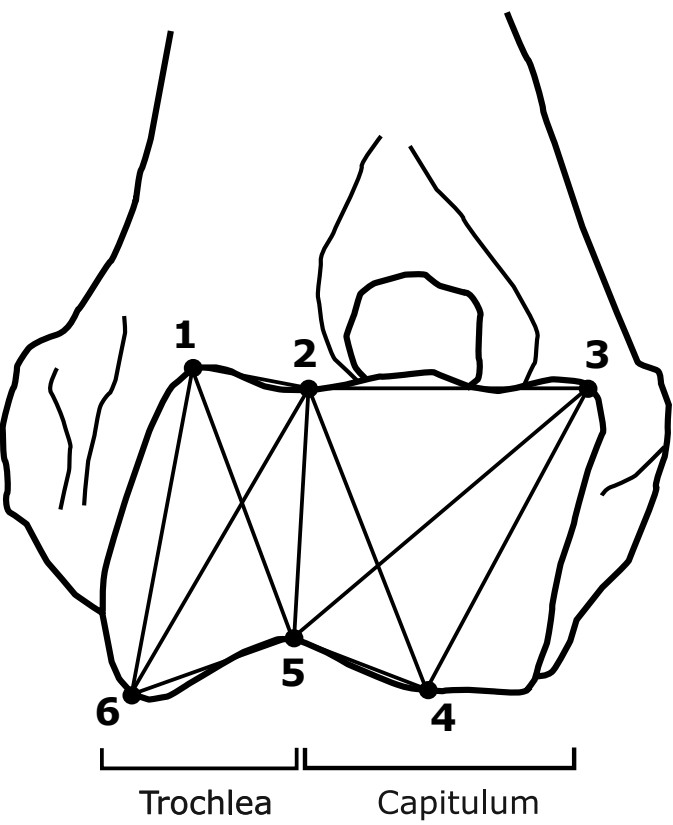

**Figure 4  Schematic representation of an elbow joint with truss coordinates and distances marked out as dots and lines respectively.** Figure adapted from *Andersson (2004)*.

## Software and graphics

All tests were carried out in R version 4.2.2 (*R Core Team, 2022*). The code and complementary files can be found in Dataset S1. Boxplots were created of the 13 indices of the linear morphometrics. Colours for plots were selected using the webpage "Coloring for colorblindness" (*Nichols, 2023*). Plots were created using ggplot2, ggtext, ggpubr and tidyverse packages (*Wickham, 2016*; *Wickham et al., 2019*; *Wilke & Wiernik, 2022*; *Kassambara, 2023*). Silhouettes of the different extant species were acquired through PhyloPic (*Keesey, 2023*).

## Permutational multivariate analysis of variance

Permutational multivariate analysis of variance (PERMANOVA) (*Anderson, 2001*) was used for analysis of the linear morphometrics. It tests if the centroids of a group of objects are the same. The test is a non-parametric alternative to multiple analysis of variance (MANOVA). It is used here because Shapiro–Wilks normality test results were found to be largely non-parametric for the linear and truss data (Table S5). The assumption for PERMANOVA is that the observations are exchangeable under the null hypothesis. Therefore, objects must be independent and have similar multivariate dispersion. A PERMANOVA can be affected by heterogeneous dispersions in an unbalanced design

(*Anderson & Walsh, 2013*; *Anderson et al., 2017*). Therefore, groups of $N = 1$ and $N = 2$ were removed from the analysis and assumption testing (Brown hyaena, Coyote and *Ictitherium ebu*).

Assumption tests for similar multivariate dispersion were carried out using the vegan package (*Oksanen et al., 2022*). A multivariate analogue of Levene's test for homogeneity of variances (betadisper) (*Anderson, 2006*) was used with Euclidean distances, after which an analysis of variance (ANOVA) (*Fisher, 1921*), as well as a permutation test of multivariate homogeneity of group dispersions (permutest) (*Legendre, Oksanen & ter Braak, 2011*) were used to validate the assumption of similar multivariate dispersion.

For species, the ANOVA yielded a p of 0.01, while the permutest yielded a p of 0.01, both of which are significant ($\alpha = 0.05$). For family both results were significant as well, with the ANOVA and permutest yielding a p of 0.004 and 0.005 respectively. Thus, multivariate dispersion was not similar, which can cause the test to be too conservative when there is large dispersion in groups of large numbers of samples and too liberal when there is large dispersion in groups of small numbers of samples (*Anderson & Walsh, 2013*).

A two-way PERMANOVA with Euclidean distances and 999 permutations was carried out to compare the indices between family and species, using the function adonis2 from the vegan package (*Oksanen et al., 2022*). Species is nested in family. Therefore the performed PERMANOVA is nested as well. *Post-hoc* Holm-corrected pairwise PERMANOVAs were carried out to identify differences in multivariate means between pairs of species using the function pairwise.adonis from the wrapper function PairwiseAdonis (*Holm, 1979*); Martinez (*Arbizu, 2020*). The Holm method of *post-hoc* correction is a more powerful sequentially rejective Bonferroni correction.

### Non-metric multidimensional scaling

Non-metric multidimensional scaling (NMDS) (*Kruskal, 1964*) with a maximum of 999 random restarts, two dimensions and Euclidean distances was carried out using the function metaMDS from the vegan package (*Oksanen et al., 2022*) for both the indices as well as the Truss distances. Its goal is to plot dissimilar objects far apart from each other and similar objects close together in ordination space (*Legendre & Legendre, 1998*). First, a distance matrix is constructed using Euclidean distances, as the data are non-ecological. A number of dimensions is chosen, in our case $k = 2$ for ease of interpretability. An initial configuration is chosen; in the case of metaMDS, this is done with metric scaling (*Oksanen et al., 2022*). This initial configuration is important, as the solution to the algorithm that is used depends partly on this configuration (*Legendre & Legendre, 1998*).

A matrix of fitted distances is calculated, then compared to the initial distances using monotone regression (which is non-metric) fitted by least-squares (*Legendre & Legendre, 1998*). Goodness of fit (stress) is used to evaluate the regression, which measures how far the new configuration is from being monotonic to the original distances. It is a relative measure, as it only measures the decrease in lack-of-fit between iterations in this procedure. The configuration is then moved slightly in the direction in which stress decreases the most rapidly (*Kruskal, 1964*; *Legendre & Legendre, 1998*). The matrix is then recalculated, and steps are repeated until a minimum lack-of-fit is reached and no more progress can be made

or until a tolerated lack-of-fit is reached (*Legendre & Legendre, 1998*). These then become the coordinates of our two-dimensional ordination. In our case, the programme is allowed to carry out up to 999 random restarts before the process is halted (*Oksanen et al., 2022*). Data are then centred, as well as rotated so that the first principal component will be on the first axis. The variable scores of the NMDS were used to explain the ordination. The NMDS results were validated using a Shepard diagram and goodness of fit of individual points (*Dexter, Rollwagen-Bollens & Bollens, 2018*), using the stressplot and goodness functions of the vegan package (*Oksanen et al., 2022*). These are available in the supplementary material (Fig. S1, Fig. S2, Table S6).

## Stepwise flexible discriminant analysis

Stepwise variable selection was carried out for species, family, locomotion and habitat on both datasets. Using a greedy Wilk's lambda F-test (*Mardia, Kent & Bibby, 1979*) from the klaR package (*Weihs et al., 2005*) the indices were selected based on an F-test decision of 0.05 (Table S7). The variable selection works by defining a start variable that separates the group most, then selects additional variables (*Weihs et al., 2005*). It makes this selection based on the Wilk's lambda criterion, which means it selects the one which minimises Wilk's lambda of the model, adding more variables if the $p$-value still shows statistical significance ($p = 0.05$).

The data were divided into 30% training data and 70% validation data using the caret and tidyverse packages (*Wickham et al., 2019*; *Kuhn, 2008*). The training data were then used to carry out flexible discriminant analysis (FDA) (*Hastie, Tibshirani & Buja, 1994*) for each variable using the function fda from the mda package (*Hastie et al., 2022*). *Ictitherium ebu*, brown hyaena and coyote were removed before dividing the data. The brown hyaena and coyote have small sample sizes ($N = 2$ and $N = 1$ respectively) and were used to validate the predictions of the data. The values for *I. ebu* were predicted to assess its similarities to other taxa and their ecologies.

Flexible discriminant analysis works as an adaptation of linear discriminant analysis (LDA) (*Hastie, Tibshirani & Buja, 1994*). Linear discriminant analysis finds a reduced number of discriminate coordinate functions to be able to optimally separate groups. This number is always the number of groups minus one. The non-parametric multiresponse regression technique BRUTO (*Hastie, 1989*) was used for FDA. It generates a very large base set automatically, then achieves parsimony by shrinking coefficients in a sensible, structured manner (*Hastie, Tibshirani & Buja, 1994*). The function "bruto" in the mda package functions by fitting a model by adaptive backfitting using smoothing splines (*Hastie et al., 2022*). The number of adaptive models is equal to the number of response variables in the model, but for each variable the same amount of smoothing is used. The variable can either be omitted, linear, or fitted by a smoothing spline. During each step of the backfitting procedure, model selection is based on an approximation of the generalised cross-validation criterion (*Hastie, Tibshirani & Buja, 1994*; *Hastie et al., 2022*). Once selection has finished, the model is backfitted using the chosen amount of smoothing (*Hastie et al., 2022*). This analysis was preferred over regular LDA, as LDA assumes normal
**Table 4** Results of the two-way permutational analysis of variance (PERMANOVA) of indices compared to species and family.

|  | Df | SumOfSqs | R2 | F | Pr(>F) |
|---|---|---|---|---|---|
| Family | 2 | 0.23 | 0.24 | 24.08 | 0.001 |
| Family:Species | 7 | 0.40 | 0.42 | 12.07 | 0.001 |
| Residual | 69 | 0.33 | 0.34 |  |  |
| Total | 78 | 0.95 | 1 |  |  |

Notes.
Df, Degrees of freedom; SumofSqs, Sum of Squares; R2, R-squared; F, F-statistic; Pr(<F), Significance.

distributions in the data. Shapiro–Wilks normality test results were found to be largely non-parametric for the linear and truss data (Table S5).

This model is then validated by testing how accurately the model predicts the validation data using the "predict" function from base-R (*R Core Team, 2022*). The variable most similar to *I. ebu*, coyote and brown hyaena was predicted in each model using the same function.

## RESULTS

### Linear morphometrics
#### *Permutational analysis of variance*
A two-way permutational analysis of variance (PERMANOVA) was carried out on the indices to compare if species and family could successfully be distinguished by their multivariate means (Table 4). Species were nested in family and 999 permutations were run. Both groupings are significant. The pseudo-F-statistic is much higher for family than for species, indicating more pronounced group separation between the two families than between individual species.

Pairwise one-way PERMANOVAs were carried out with *post-hoc* Holm corrections to assess how different the extant species are from each other, as they are from just two families. The results revealsignificant differences between some of the species (Table S8). Boxplots of the 13 indices were generated to visually compare the different species (Fig. S3). The indices show the highest number of significant differences between the maned wolf and the other species (50 significant indices). The side striped jackal shows the lowest number of significant differences (19 significant indices). The indices HEI (32), BI (28), and MCHUM (26) provide the highest number of significant differences (Fig. S3, Table 3, Table S8), while URI (16), GI (12), and HRI (eight) provide the lowest number of significant differences.

While *Ictitherium ebu* has no significant pairwise comparisons due to only being a single individual, it can be visually separated from other species in the boxplots (Fig. S3). The animal has lower than average values of OLI, GI, RRI, HEI, FEI, HFI and a high BI (Table 3). Except for GI and HFI, these indices are in the range of those of the maned wolf. The maned wolf and *I. ebu* also show overlap in SMI, OLI, URI, MCHUM and MCRI (Table 3).

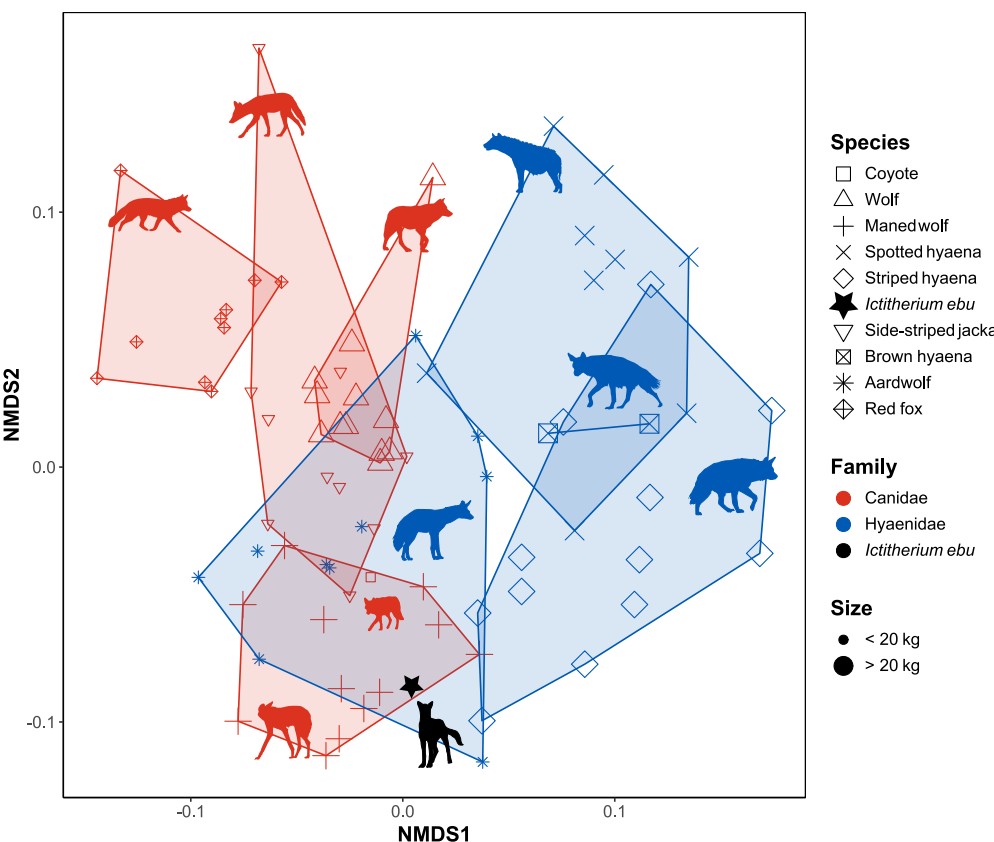

**Figure 5 NMDS plot of the morphometric indices.** Species are labelled by shape, family by colour and size by largeness of the dots. Minimum convex polygons for each species are shown in the colour of their family. *I. ebu* is plotted as a black star. NMDS1 is inverted to plot smaller species on the left and larger species on the right. Silhouette of *Ictitherium ebu* traced from the reconstruction by Javier Herbozo. Extant species silhouettes from Phylopic (*Keesey, 2023*): Aardwolf: https://www.phylopic.org/images/e1072c6c-39fb-4375-aec5-dc0405ed1e3f/proteles-cristatus. Brown hyaena: https://www.phylopic.org/images/55b1ad9b-93d5-4fa9-83a5-4231c07c8620/parahyaena-brunnea. Coyote: https://www.phylopic.org/images/5a0398e3-a455-4ca6-ba86-cf3f1b25977a/canis-latrans. Fox: https://www.phylopic.org/images/9b3e3567-2238-40bf-ab5d-bcb6ce7b32d5/vulpes-vulpes. Maned wolf: https://www.phylopic.org/images/72e1b4c0-5df9-47a2-9950-253fac62080f/chrysocyon-brachyurus. Side-striped jackal: https://www.phylopic.org/images/2bbc62f3-696f-4ac1-a235-6aeff17d2a88/lupulella-adusta. Spotted hyaena: https://www.phylopic.org/images/f1b665ae-8fe9-42e4-b03a-4e9ae8213244/crocuta-crocuta. Striped hyaena: https://www.phylopic.org/images/efd94e33-3966-4808-af47-3c58b333f9af/hyaena-hyaena. Wolf: https://www.phylopic.org/images/8cad2b22-30d3-4cbd-86a3-a6d2d004b201/canis-lupus[p].

### Non-metric multidimensional scaling

A convergent solution was found after 20 tries by the model. Stress for the non-metric multidimensional scaling is 0.13. The Shepards plot shows an $R^2$ of the non-metric fit of 0.98 (Fig. S1). In the goodness of fit table (Table S6), none of the values of goodness of fit are above 0.05.

Axis 1 of the NMDS is primarily controlled by SMI and HEI (Tables 3 and 5). SMI represents the muscles acting across the shoulder joint while HEI represents the relative

**Table 5 NMDS1 and NMDS2 loadings of the non-metric multidimensional scaling of the morphometric indices.**

|  | NMDS1 | NMDS2 |
|---|---|---|
| SMI | −0.16 | 0.08 |
| BI | −0.01 | −0.06 |
| HRI | −0.05 | 0.03 |
| OLI | 0.03 | 0.17 |
| URI | −0.02 | 0.17 |
| FRI | −0.03 | 0.04 |
| GI | −0.03 | 0.05 |
| FEI | −0.02 | 0.06 |
| HEI | −0.12 | 0.04 |
| RRI | 0.01 | 0.12 |
| MCRI | 0.06 | −0.0004 |
| HFI | 0.08 | −0.01 |
| MCHUM | −0.01 | −0.05 |

area for the origins of the forearm flexors, pronators, and supinators. Canids plot more on the left side and hyaenids more on the right side Fig. 5.

Axis 2 is primarily controlled by OLI, URI and RRI (Tables 3 and 5). OLI relates to the muscles used in elbow extension, while RRI is the indicator for radial robustness and stress resistance. URI represents the overall robustness and resistance to stress of the ulna. All these indices relate to the robustness of the forearm and extension of the elbow. The maned wolf, aardwolf and striped hyaena plot low on this axis, while the red fox, side-striped jackal and spotted hyaena plot high. While the aardwolf overlaps all Canidae except the red fox, the other Hyaenidae do not show overlap with the Canidae. The aardwolf and maned wolf, which are the two non-cursorial species, plot close together. *I. ebu* plots in the polygon formed by the maned wolf as well as the aardwolf.

## Flexible discriminant analysis

Stepwise flexible discriminant analysis of the different species was performed on the 13 morphometric indices, of which 10 were selected using a greedy Wilk's lambda F-test. The indices FEI, BI, HFI, MCRI, GI, OLI, MCHUM, URI, HEI and FRI (Table 3) were selected based on an F-test decision of 0.05 (Table 3, Table S7). The resulting plot of the first two axes mainly separates the fox and maned wolf from the other data (Fig. 6).

Together, the first two dimensions account for 71.63% of the variance of the data. The first dimension accounts for 44.56% of the variance, while the second dimension accounts for 27.06% of the variance. The test data were predicted by the model with an accuracy of 0.9.

The red fox is separated from the rest of the data and clusters to the left (Fig. 6). The more cursorial Hyaenidae plot together on the right. In the centre are the other Canidae and the aardwolf, with the maned wolf plotting lower on Canonical variate 2 (CV2). The coyote is predicted to be a wolf, while the brown hyaena data is predicted to be from a

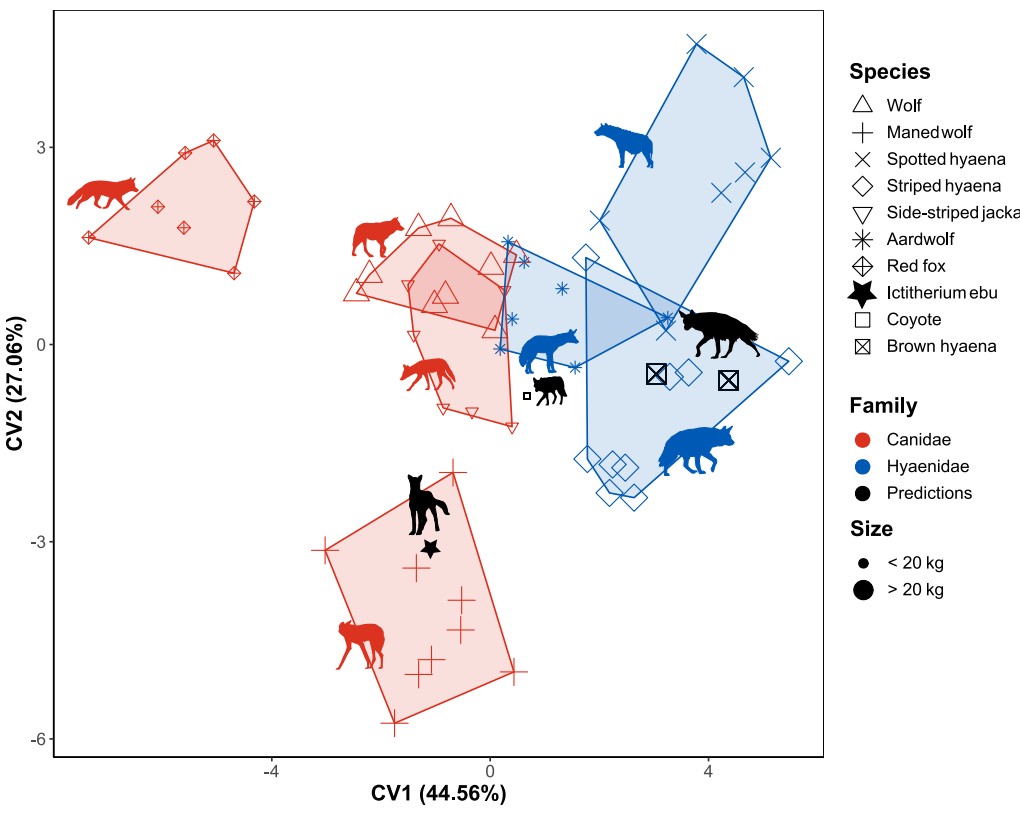

**Figure 6** **Flexible discriminant analysis of the morphometric indices.** Species are labelled by shape, family by colour and size by largeness of the dots. Minimum convex polygons for each species are shown in the colour of their family. The predicted value of *I. ebu* is plotted as a black star. CV1 is inverted to plot smaller species on the left and larger species on the right. Silhouette of *Ictitherium ebu* traced from the reconstruction by Javier Herbozo. Extant species silhouettes from Phylopic (*Keesey, 2023*): Aardwolf: https://www.phylopic.org/images/e1072c6c-39fb-4375-aec5-dc0405ed1e3f/proteles-cristatus. Brown hyaena: https://www.phylopic.org/images/55b1ad9b-93d5-4fa9-83a5-4231c07c8620/parahyaena-brunnea. Coyote: https://www.phylopic.org/images/5a0398e3-a455-4ca6-ba86-cf3f1b25977a/canis-latrans. Fox: https://www.phylopic.org/images/9b3e3567-2238-40bf-ab5d-bcb6ce7b32d5/vulpes-vulpes. Maned wolf: https://www.phylopic.org/images/72e1b4c0-5df9-47a2-9950-253fac62080f/chrysocyon-brachyurus. Side-striped jackal: https://www.phylopic.org/images/2bbc62f3-696f-4ac1-a235-6aeff17d2a88/lupulella-adusta. Spotted hyaena: https://www.phylopic.org/images/f1b665ae-8fe9-42e4-b03a-4e9ae8213244/crocuta-crocuta. Striped hyaena: https://www.phylopic.org/images/efd94e33-3966-4808-af47-3c58b333f9af/hyaena-hyaena. Wolf: https://www.phylopic.org/images/8cad2b22-30d3-4cbd-86a3-a6d2d004b201/canis-lupus.

striped hyaena. When *I. ebu* is added to the model, it is predicted to be a maned wolf specimen.

Three other models were generated based on family, locomotion and habitat. As FDA reduces the dimensions of the tested groups by one, these were one-dimensional models (Fig. S4). The family model based on the training data reclassified the test data with an accuracy of 0.95, based on the indices HEI, MCHUM, HFI, GI, SMI, FRI and URI (Table 3, Table S7). The extant specimens were correctly classified. *I. ebu* scores a low value on CV1 and was predicted as Canidae, which is incorrect. The locomotion model reclassified the test data with an accuracy of 0.86, based on OLI, MCRI and HFI (Table 3, Table S7).

**Table 6   NMDS1 and NMDS2 loadings of the non-metric multidimensional scaling for the truss analysis.**

| Distances | NMDS1 | NMDS2 |
|---|---|---|
| 2-6 | 0.10 | 0.01 |
| 2-5 | −1.53 | −0.45 |
| 3-5 | −0.03 | −0.02 |
| 1-5 | −0.02 | −0.18 |
| 2-4 | 0.36 | −0.18 |
| 5-6 | 3.43 | 0.71 |
| 4-5 | 3.12 | 0.07 |
| 3-4 | −1.04 | −0.11 |
| 2-3 | 2.15 | 0.37 |
| 1-2 | −0.65 | −0.02 |
| 1-6 | 0.44 | −0.06 |

All extant specimens are predicted to be cursorial, which is correct. *I. ebu* was predicted as cursorial and has an intermediate value on CV1. Finally, the habitat model reclassified the test data with an accuracy of 0.95, based on BI, HFI, FEI, MCRI and MCHUM (Table 3, Table S7). The coyote is correctly predicted to be a mixed habitat species, while one of the brown hyaenas is incorrectly also assigned to being mixed habitat. *I. ebu* was predicted as an open habitat species and has a value slightly above intermediate for the plot.

## Truss analysis
### *Non-metric multidimensional scaling*
A repeat of the best solution was reached after 26 tries. Stress is 0.03. The Shepards plot shows an $R^2$ of the non-metric fit of 0.99 (Fig. S2) and no goodness of fit value is higher than 0.01 for the individual points (Table S6). There is a lot of overlap between species, mainly on the left side of the plot (Fig. 7).

The first axis is mostly controlled by distance 5–6, the distal width of the trochlea, and distance 4–5, the distal width of the capitulum (Table 6) and reflects overall size. The second axis is primarily controlled by distance 5–6 and distance 2–5 and relates to the distal extension of the trochlea. It separates species like the wolf, with a less extended trochlea and squarer anterior distal humerus, from species like the striped hyaena, with a more extended trochlea. *I. ebu* is separated from the other species, indicating an intermediate size and trochlear extension.

## Flexible discriminant analysis
Flexible discriminant analysis of the different species was carried out for the different truss distances. The model accounts for 90.19% of the variance in its first two dimensions. CV1 accounts for 51.91% of the variance. It reflects the shape of the capitulum and with it, the overall anterior distal humerus from a squarer (wolf) to a more rectangular (brown hyaena) shape. CV2 accounts for 38.28% of the variance. It reflects the overall size of the specimens (Fig. 8). Model accuracy is 0.83. It separates species in a manner similar to the NMDS. The coyote is predicted to be an aardwolf, while the brown hyaena data is closest to

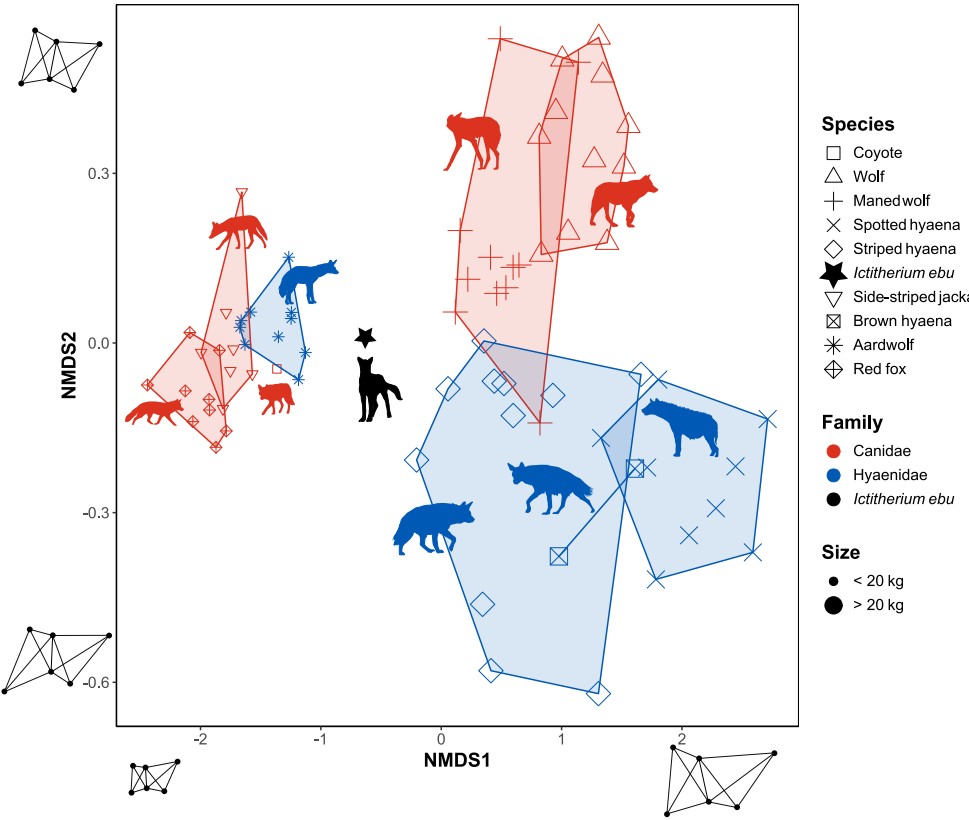

**Figure 7   NMDS plot of the truss analysis.** Species are labelled by shape and family by colour. Minimum convex polygons for each species are shown in the colour of their family. The predicted value of *I. ebu* is plotted as a black star. NMDS2 is plotted in reverse to be able to more easily compare it to the linear NMDS. Silhouette of *Ictitherium ebu* traced from the reconstruction by Javier Herbozo. Extant species silhouettes from Phylopic (*Keesey, 2023*): Aardwolf: https://www.phylopic.org/images/e1072c6c-39fb-4375-aec5-dc0405ed1e3f/proteles-cristatus. Brown hyaena: https://www.phylopic.org/images/55b1ad9b-93d5-4fa9-83a5-4231c07c8620/parahyaena-brunnea. Coyote: https://www.phylopic.org/images/5a0398e3-a455-4ca6-ba86-cf3f1b25977a/canis-latrans. Fox: https://www.phylopic.org/images/9b3e3567-2238-40bf-ab5d-bcb6ce7b32d5/vulpes-vulpes. Maned wolf: https://www.phylopic.org/images/72e1b4c0-5df9-47a2-9950-253fac62080f/chrysocyon-brachyurus. Side-striped jackal: https://www.phylopic.org/images/2bbc62f3-696f-4ac1-a235-6aeff17d2a88/lupulella-adusta. Spotted hyaena: https://www.phylopic.org/images/f1b665ae-8fe9-42e4-b03a-4e9ae8213244/crocuta-crocuta. Striped hyaena: https://www.phylopic.org/images/efd94e33-3966-4808-af47-3c58b333f9af/hyaena-hyaena. Wolf: https://www.phylopic.org/images/8cad2b22-30d3-4cbd-86a3-a6d2d004b201/canis-lupus.

the striped hyaena. When *I. ebu* is added to the model, it is predicted as being a specimen of the aardwolf.

Three other models were generated based on family, locomotion and habitat. As FDA reduces dimensions of the tested groups by one, these were one-dimensional models (Fig. S5). The family model reclassified the test data with an accuracy of 0.8. *I. ebu* was predicted as Hyaenidae and has a relatively high value on the plot. The extant species are correctly assigned to their own families. The locomotion model reclassified the test data with an accuracy of 0.75. *I. ebu* has a very high value and was predicted as having a generalist

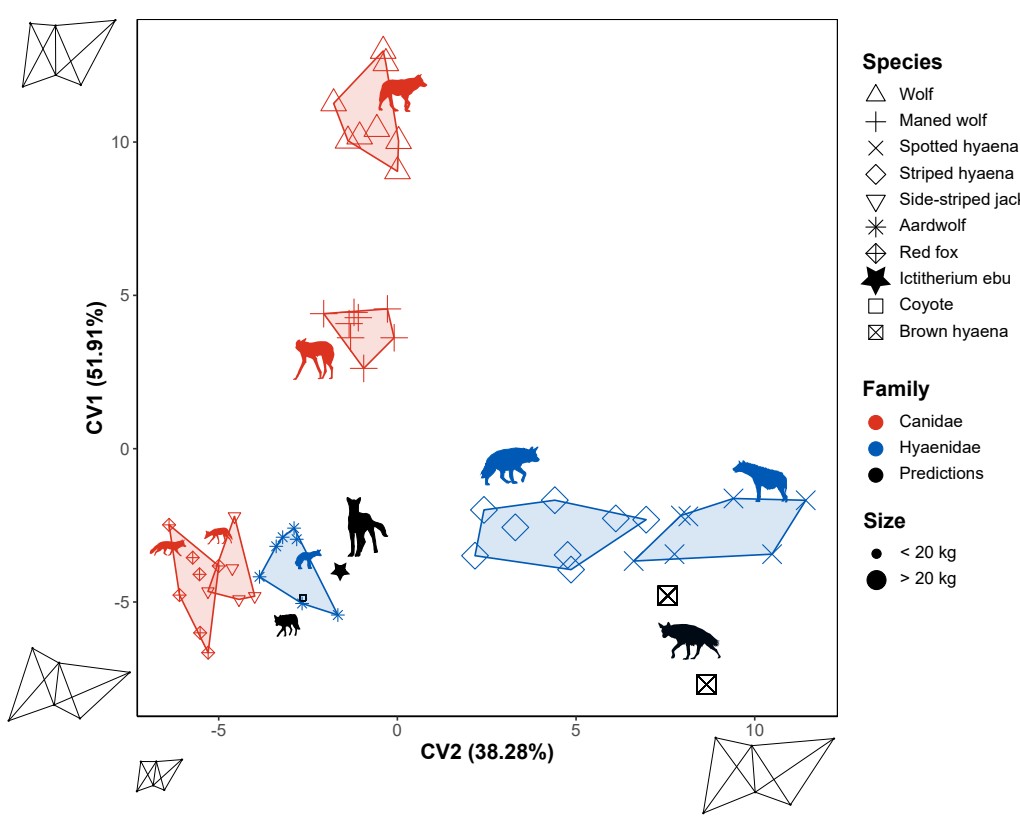

**Figure 8** **Flexible discriminant analysis of the truss distances.** Species are labelled by shape and family by colour. Minimum convex polygons for each species are shown in the colour of their family. The predicted value of *I. ebu* is plotted as a black dot. Axis 2 was plotted horizontally and axis 1 vertically, with axis 2 reversed for ease of comparison to the other plots. Silhouette of *Ictitherium ebu* traced from the reconstruction by Javier Herbozo. Extant species silhouettes from Phylopic (*Keesey, 2023*): Aardwolf: https://www.phylopic.org/images/e1072c6c-39fb-4375-aec5-dc0405ed1e3f/proteles-cristatus. Brown hyaena: https://www.phylopic.org/images/55b1ad9b-93d5-4fa9-83a5-4231c07c8620/parahyaena-brunnea. Coyote: https://www.phylopic.org/images/5a0398e3-a455-4ca6-ba86-cf3f1b25977a/canis-latrans. Fox: https://www.phylopic.org/images/9b3e3567-2238-40bf-ab5d-bcb6ce7b32d5/vulpes-vulpes. Maned wolf: https://www.phylopic.org/images/72e1b4c0-5df9-47a2-9950-253fac62080f/chrysocyon-brachyurus. Side-striped jackal: https://www.phylopic.org/images/2bbc62f3-696f-4ac1-a235-6aeff17d2a88/lupulella-adusta. Spotted hyaena: https://www.phylopic.org/images/f1b665ae-8fe9-42e4-b03a-4e9ae8213244/crocuta-crocuta. Striped hyaena: https://www.phylopic.org/images/efd94e33-3966-4808-af47-3c58b333f9af/hyaena-hyaena. Wolf: https://www.phylopic.org/images/8cad2b22-30d3-4cbd-86a3-a6d2d004b201/canis-lupus.

locomotor mode. The coyote was also assigned a generalist category, while the brown hyaena was once predicted to be cursorial and once a generalist. Finally, the habitat model reclassified the test data with an accuracy of 0.47 and *I. ebu* was predicted as an open habitat species with a value close to the mean. The extant species were also predicted to be open habitat species, which is incorrect for the coyote, but correct for the brown hyaena.
## DISCUSSION

### Flexible discriminant analysis validation

The flexible discriminant analyses of species have high reclassification rates and sensibly interpret the brown hyaena and coyote. The brown hyaena is classified as the striped hyaena in both the linear and truss flexible discriminant analysis. This classification makes sense, as they overlap in the NMDS of both methods and are simply put, large species of hyaena (Figs. 5 and 7). The coyote is predicted by its overall morphology to be a wolf, another member of its genus. The truss analysis would instead predict an aardwolf, as it is small like *I. ebu.*

The family models appear quite accurate as well, with no classification errors and high reclassification rates in both the linear morphometric and truss analysis (0.95 and 0.8 respectively). The incorrect predictions of the coyote and brown hyaena show that the locomotor and habitat truss models are less accurate (Fig. S5). In the habitat model of the truss analysis the brown hyaena even appears to be a far outlier, indicating that its morphology is not captured well by the model. This model also has the lowest reclassification score, at 0.47.

### Permutational analysis of variance

Indices HEI, BI and MCHUM (Table 3) have the most significant differences in the pairwise PERMANOVAs (Fig. S3), showing that forelimb length ratios and epicondylar breadth are important characteristics when separating species. HEI and BI are also important characteristics for separating cursorial carnivores from other groups (*Samuels, Meachen & Sakai, 2013*), indicating that these indices detect similar differences in morphology in this smaller dataset. The similar proportions of *I. ebu* and the maned wolf would suggest similar adaptations.

### Size differences

Size explains a large amount of dissimilarity in all of the species plots (Figs. 5, 6, 7 and 8). This pattern is particularly clear in the truss plots (Figs. 7 and 8), where the first axis of the NMDS and the second axis of the Truss seem to show mostly size differences. A similar size pattern was observed by *Andersson (2004)*, Fig. 6), where small species are quite uniform in the main axis that reflects shape. At larger sizes, the differences between families increase with size. These dissimilarities may relate to a shift in diet from small to large prey (*Carbone et al., 1999*; *Andersson, 2004*).

### Shape differences

The linear morphometrics NMDS results show that forearm robustness and the relative size of the olecranon present important differences in shape in these species (Fig. 5). The maned wolf, aardwolf and striped hyaena all plot lower on NMDS2. They all have a shorter olecranon, which reflects the fact that these predators do not need to apply much strength with their forelimbs during hunting and handling of prey (*Martín-Serra, Figueirido & Palmqvist, 2016*). Furthermore, they have less robust forearms. The striped hyaena is predominantly a scavenger (*Rieger, 1981*) and thus plots lower than the spotted

hyaena, which more commonly hunts large prey (*Hayssen & Noonan, 2021*). The maned wolf is a generalist species that feeds on vegetal items as well as small to medium-sized prey (*Rodrigues et al., 2007*). The aardwolf uses its long tongue to consume termites, but never uses its front legs to dig (*Kruuk & Sands, 1972*). Both species have a generalist locomotor mode (*Samuels, Meachen & Sakai, 2013*). *I. ebu* plots within the range of the maned wolf, not far from the aardwolf, indicating some overlap with these species in forearm function. Overall, it appears that NMDS2 reflects shape that is impacted by hunting strategy and cursoriality.

The truss analysis NMDS shows that for the elbow joint the extension of the medial trochlear flanges is an important discriminator (Fig. 7). In this analysis, Hyaenidae have more extended medial trochlear flanges. This extension reflects an increase in lateral stability during humeral articulation, which is an indicator of an animal that grapples with its prey (*Andersson, 2004*). *Andersson (2004)* placed both Hyaenidae and Canidae in a group with the cheetah (*Acinonyx jubatus*) to indicate non-grapplers. Only the maned wolf is interpreted as a grappler but has the elbow joint morphology of a non-grappler (*Andersson, 2004*). However, as is the case for Fig. 6 of *Andersson (2004)*, the canids plot higher than the hyaenids.

## Phylogenetic signal

The linear morphometrics NMDS and FDA species plots both show a strong separation between families (Figs. 5 and 6). These differences may be due to size, as most hyaenids are larger than canids, or due to a phylogenetic signal in the differences in morphology, such as the SMI and HEI (Table 3) of the NMDS (Fig. 5).

The presence of a phylogenetic signal within the shape variables cannot be eliminated because extant canids have a cursorial ancestry (*Andersson, 2004*). This can be seen in the maned wolf, a generalist species that has traits of a more cursorial animal (*Andersson, 2004*; *Samuels, Meachen & Sakai, 2013*). A phylogenetic signal might even be more widely present in the data. However, skeletons not only reflect relatedness, but their form also reflects their function. Therefore, while phylogenetic corrections could have been implemented, they would be of limited use in this relatively simple, two-family case.

## The ecomorphology of *I. ebu*

While a quick look at the variance showed a closer similarity of *I. ebu* to the maned wolf in the overall morphology of the limb bones (Fig. S3), both the maned wolf and aardwolf showed similar values in the NMDS (Figs. 5 and 7). The linear species FDA (Fig. 6) predicted *I. ebu* to be similar to the maned wolf, far from the other species, in accordance with the interpretation of the boxplots. The truss analysis of the distal humerus showed a closer affinity to the aardwolf in both the NMDS and species FDA (Fig. 7, Fig. 8), which is likely due to its size, which appears to correspond with a more uniform elbow joint shape (*Andersson, 2004*).

While the family of *I. ebu* was predicted as Canidae in the linear analysis, the truss analysis interpreted the family as Hyaenidae (Fig. S4, Fig. S5). It may be that the morphology of *I. ebu* is overall more similar to Canidae, but the elbow is more similar to Hyaenidae.

While the locomotor model predicts cursorial locomotion for the overall morphology, the elbow predicts generalist locomotion. However, both species predictions would suggest a generalist locomotor mode. The models interpret *I. ebu* as an open habitat animal, in agreement with *Werdelin (2003)*. The Lower Nawata represents a relatively mixed habitat, as pure grassland was likely not long-lived but could have been present (*Wynn, 2003*). The maned wolf, while mainly occurring in grasslands, can also be found in savannah woodlands (*Queirolo et al., 2011*). Given its similar morphology to the maned wolf, *I. ebu* could also have been present in these types of environments.

Overall, the long, gracile limbs of *I. ebu* were not an adaptation for cursoriality, but for being able to look over the tall grasses of its environment and pounce on prey, similar to the maned wolf (*Hildebrand, 1954*; *Janis & Wilhelm, 1993*) and possibly the serval (*Leptailurus serval*), a felid not analysed here that removes prey from crevices (*Janis & Wilhelm, 1993*; *Ewer, 1998*). The longer legs of *I. ebu* also contribute to its walking efficiency, as locomotion efficiency increases with longer legs at all gaits (*Pennycuick, 1975*; *Janis & Wilhelm, 1993*).

## CONCLUSIONS

*Ictitherium ebu* was hypothesised to be cursorial, based on its long, gracile limbs. Through a combination of two and three dimensional morphometric techniques it was found that *I. ebu* resembled the maned wolf in the overall morphology of the limbs, while it resembled the aardwolf in the morphology of the knee joint. As neither of these animals is cursorial, *I. ebu* would not have been cursorial either. Similar to the maned wolf, the long slender limbs of *I. ebu* would have been an adaptation for looking over the tall grasses of its environment, pouncing on prey and walking efficiency. Further research on the ecomorphology of the hyaenids of Lothagam and other Late Miocene African sites will help to categorise the as yet understudied African community patterns of Hyaenidae.

## ACKNOWLEDGEMENTS

We would like to thank Franziska "Poppy" Pezzei for her suggestion of applying for the Otterborg grant and Thorben Schöfisch for his help with the application. We would like to thank Chun Hei Ho for allowing us to borrow his camera equipment and for patient guidance in using his equipment. Thanks to Christiane Funk and Doreen Bayer of the Museum för Naturkunde, Katrin Krohmann and Irina Ruf of the Senckenberg Naturmuseum, Jan Decher and Christian Montermann of the Alexander Koenig Zoological Research Museum, Emmanuel Gilissen of the Royal Museum for Central Africa, Pepijn Kamminga of Naturalis Biodiversity Center, Paolo Agnelli of La Specola and Daniela Kalthoff of Naturhistoriska Riksmuseet for allowing us to visit their collections. Your guidance, help and answers to our questions were much appreciated. The manuscript was greatly improved with the help of editor Michela Johnson, an anonymous reviewer, Juan Antonio Pérez-Claros, P. David Polly and Charles J. Salcido.

### Funding

Julien van der Hoek received funding from the Otterborg Scholarship fund. The funders had no role in study design, data collection and analysis, decision to publish, or preparation of the manuscript.

### Grant Disclosures

The following grant information was disclosed by the authors:
The Otterborg Scholarship fund.

### Competing Interests

The authors declare there are no competing interests.

### Author Contributions

- Julien van der Hoek conceived and designed the experiments, performed the experiments, analyzed the data, prepared figures and/or tables, authored or reviewed drafts of the article, and approved the final draft.
- Lars Werdelin conceived and designed the experiments, analyzed the data, authored or reviewed drafts of the article, and approved the final draft.

### Data Availability

The raw data and code are available in the Supplemental File.

### Supplemental Information

Supplemental information for this article can be found online at http://dx.doi.org/10.7717/peerj.17405#supplemental-information.

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
