# Peer review of "A hyaena on stilts: comparison of the limb morphology of Ictitherium ebu (Mammalia: Hyaenidae) from the Late Miocene of Lothagam, Turkana Basin, Kenya with extant Canidae and Hyaenidae"

_PeerJ, doi:10.7717/peerj.17405_

## Round 0.1 · original submission · Minor Revisions

A well-written, original, clear, and interesting piece of work, and I congratulate the authors on their research findings. Make sure to carefully read through all reviewers' suggestions/comments and double-check the categories used for locomotion as explained by Reviewer 2.

Reviewer 1 ·

Basic reporting

Writing is clear and unambiguous; usage of English is professional throughout. The literature references are more than sufficient--through them, I found some papers about which I had not known previously; much appreciation to the authors. Article is professionally structured, including the figures and tables; all raw data appear to be shared. The study is self-contained with relevant results to the hypothesis.

I have two suggestions to improve reading comprehension:

(a) In the Results: the acronyms for the linear-morphometric indices tend to interrupt one's reading so that the reader can look up the acronyms. Ideally one would write out the whole term (e.g., "Humeral Epicondylar Index (HEI)"); but, with this many index acronyms in that section, writing out everything may be bulky as well. Perhaps a solution can be a reference to Table 3 with each mention of an index acronym (e.g., "HEI, [...] HRI (Table 3)"). Right now, the only reference in lines 309-310 is to a figure, which is good, but it also needs a reference to Table 3.

(b) I would appreciate more internal references to figures and tables, particularly in the Discussion. I had to remind myself which figure the text at any point might be discussing.

Experimental design

The original primary research is within PeerJ's Aims and Scope. The authors clearly define their research question. The investigation is rigorous.

The Methods are described with sufficient detail and information to replicate. The section is impressively thorough. I look forward to being able to share the published paper with students, as this Methods section is an excellent example for them to emulate.

One of the paper's primary results is that "multivariate statistical analysis primarily separated the data based on taxonomy". Accordingly, the authors write that, in some analyses, Axis 1 is defined by family while Axis 2 appears to represent size. This is interesting because Axis 1 is usually the size axis: size is expected to be the most significant driver of differences among samples. A phylogenetic signal is clearly present (also acknowledged in passing by the authors), which makes me wonder how the results might differ with phylogenetic comparative methods (PCM). Perhaps a phylogeneticist would advise major revisions in order to implement PCM. But... regardless of relatedness, an animal's skeletal morphology functions a certain way--its form determines its function. So I would advise the authors to add only an explanation of this consideration--demonstrating their awareness of possible phylogenetic bias, but explaining that phylogenetic corrections are of limited use in this (relatively simple, two-family) case.

Validity of the findings

Findings are complex, suggesting a mosaic of ecomorphological function in I. ebu: overall similar to maned wolf, but with elbow joint similar to aardwolf.

In Results, under "Flexible discriminant analysis": "the first two dimensions account for 69.96% of the data". Do you mean that they account for 69.96% of the variance in the data? This would be good to re-state accurately. (Similarly: "first dimension accounts for 46.72% of the data".)

While the findings appear valid, the Discussion of the Results can be improved. As currently written, the Discussion mainly rehashes the Results--so it is repetitive. The section titled "Implications of the linear morphometrics" especially can benefit from being shortened to about half its current length.

As well, a figure comment: Figures 8 and 11, which have species on the y-axis, need clarity. I understand that each horizontal "line" of points on the strip-chart represents a different species--but, the points are jittered and therefore some end up closer to others. It would be helpful to have alternating light-grey and white horizontal bars providing a background pattern on your plot: e.g., red fox points would be in a light-grey bar, aardwolf would be in a white bar, brown hyaena in light-grey, etc. This way, we can reduce the reader getting cross-eyed when trying to understand these plots.

Additional comments

Thanks to the authors for a nicely executed analysis on a beautifully preserved set of fossil specimens.

·

Basic reporting

---Clear and unambiguous, professional English used throughout:
Yes.

---Literature references, sufficient field background/context provided:

Relevant prior literature is appropriately referenced but in my opinion, the introduction could be expanded slightly in order to better contextualize the reported findings (see the annotated ms.).
There is another aspect that I think it is appropriate to comment on here. The authors take as good some hypotheses put forward in the literature on locomotion categories that are controversial. Not everything published should be accepted uncritically. This is especially relevant given that some of the techniques used (e.g. discriminant analyses or Permanovas) and the commentary on the results obtained are based on a priori defined categories such as being cursorial or not. The results obtained by the authors are better than their statistical tests show...it is the categories that are poorly defined (see below for the details and references that support my statement).

---Professional article structure, figures, tables. Raw data shared:

All data used and results are well in the text or in the supplementary material.
The nominal structure of the article conforms to an acceptable format. However, part of the contents of certain sections perhaps should be moved:

1- Part of the contents of the discussion section are actually results. Perhaps both could be regrouped in a section called Results and Discussion. In any case it is necessary to synthesise them appreciably so as not to be repetitive.

2-The “limitation of the research” section I think it is more appropriate to move it to material and methods. In fact, this section talks about the nature of the sample and statistical questions

The supplementary information is confusing. There are too many files. Some supplementary results could be grouped in a single file, for example the two Supplemental box plots. Within each supplementary figure it would be a good idea to include a figure caption with the definitions of the abbreviations (variables and species), otherwise it is necessary to refer to the table with the definitions to interpret figures or other tables (the names of indexes are specially difficult to remember). Really it takes nothing…the space is not a problem in the supplementary material. All complementary results within a block of results (e.g. Linnear morphometric: Permanova,….etc.) could be grouped in a single file.
Some elements of the supplementary material are difficult to understand. For example to test of the validity of the photographs and removal of invalid photographs the author indicate to see: Article S1, Dataset S1, Table S3, Table S4 and Fig. S1. It is confusing. By the way, in Fig. S1, the symbols in the legend do not correspond to those in the plot. In addition such information on the validity of the photographs is not essential for the main topic analysed here and could be summarized or even removed.
In short... supplementary information should be thoroughly reviewed, synthesised and exclude those elements that are not essential to the main theme of the article.

---Self-contained with relevant results to hypotheses:

The hypotheses to be tested relate to the ecomorphology of I. ebu. Most analyses are focused to resolving this question.
However, the authors do not explain why they use Flexible discriminant analysis to assign I. ebu to a living species and family...both the species and family of I. ebu are known..it is not necessary to identify a material whose identification is known. Nor is it explained why Permanovas are used to compare current species. To assess how similar I. ebu is to current species, multidimensional scaling or a simple principal component analysis can be used. Since the relationship of these analyses to the ecomorphology of I. ebu is not direct, the authors should explain for what purpose such analyses are used.

Experimental design

-This work is original primary research and within aims and scope of PeerJ.

-The research question is clearly defined: to analyse the ecomorphology of I. ebu which is relevant and meaningful by two reasons: the postcranial of ictitheres is rarely preserved (and consequently, it is very unknow) and in this case the postcanial is clearly an evolutionary convergence with the maned wolf as authors indicate. Consequently, this research fills an identified knowledge gap.
-In general, the investigation is conducted rigorously and to a high technical standard with the exception of some analyses performed with a low number of observatiosn (I will come back to this below).

- A large percentage of the work is devoted to explaining the statistical techniques used. Such techniques are not unknown at all but they are not widely used. It is therefore reasonable to devote a certain percentage of the text to explaining them in order to be understood by a wider audience. However, although it is not strictly necessary, I do miss that the authors justify why Flexible discriminant analysis is used instead of Linear discriminant analysis or why Non-metric multidimensional scaling is used instead of metric multidimensional scaling (i.e., Principal coordinates). If the reason is that the results are more easily interpretable, it should be clearly indicated.
Methods are described with sufficient detail and all the information to replicate is supplied. Even the R-code for the analyses is supplied.

- In the case of the non-metric multidimensional scaling the use of samples of N=1 or N=2 is not a problem, but in the case of PERMANOVA is an obvious abuse of the method. I have programmed the algorithm in Wolfram Mathematica. The mathematical algorithm after doing the permutations does not produce a fatal error (division by zero or something similar). That's why the programs produce an output irrespectively of samples sizes...but it doesn't make statistical sense to permute a single observation. It can be seen as an extreme case of unbalanced design where one of the sets has a minimum sample size or (N=1). The observation for C. latrans, the two P. brunnea and the specimen of I. ebu must be excluded from those permanovas to test hypotheses where such observations are integrated within a larger set (as for example type of locomotion).
In the case of discriminant analyses the C. latrans and the two P. brunnea can be used to test the effectiveness of the analyzes by excluding them from the analysis phase and including them in the classification phase to contrast whether the prediction of the analyses predicts their ecology and locomotion.

Validity of the findings

All underlying data have been provided.
Results are robust and statistically sound with the exception of the points I have commented above.
The conclusions are appropriately stated and connected to the original question investigated (with the exception of the discriminant function for species and family at least without an additional justification). Obvious some results are clearer and others less conclusive but the general conclusion drawn from the analyses as a whole that I. ebu is most similar to the maned wolf and aardwolf is correct in my opinion.
However, I believe that there is a basic error that has caused problems for the authors in making a correct interpretation of part of their results: the categories used for locomotion.
In the context of this research to be cursorial is meaningless. In fact, in according to Samuels, Meachen & Sakai (2013) the definition of terrestrial and cursorial are:

Terrestrial: Rarely swims or climbs, may dig to make a burrow (but not extensively) (e.g., ermine, grisons).

Cursorial: Regularly displays rapid locomotion with bounding characterized by unsupported intervals (e.g., cheetah, gray wolf).

Please note that the gray wolf rarely swims or climbs, may dig to make a burrow and consequently it would be a terrestrial. Many “cursorial” canids and hyenas do not run regularly but only occasionally. Please, note as well that even a brown bear or a black bear can run at almost 50 Km/h [Garland, T. (1983). Journal of Zoology, 199, 157-170]. In other words, only a high speed without any other type of qualification does not characterize a cursor.
In one of my works, my collaborators and I briefly reflected on this issue (Martín-Serra, A., Figueirido, B., Pérez-Claros, J. A., & Palmqvist, P. (2015). Patterns of morphological integration in the appendicular skeleton of mammalian carnivores. Evolution, 69(2), 321-340.)
In such work, after a brief review of this concept, we finally conclude that: “In any case, we use the term cursor in a broader sense, by including in this category all carnivorans whose forelimbs are used primarily for terrestrial locomotion (i.e., many canids, hyaenids, and the cheetah among felids; see Table 1 for references), whereas the noncursor category includes (1) ambushing carnivorans that grapple with prey (i.e., all felids except the cheetah; see Table 1); (2) species with digging, climbing, or swimming abilities (e.g., the European badger, the raccoon, or the northern river otter)…”, i.e., all the species analysed by the authors here are cursorials.
However, Incidentally the authors have brought together Proteles and Chrysocyon within generalist locomotion...which is a natural group! but not of locomotion but of foraging behaviour: both explore the terrain for insects (in the case of the maned wolf, also for fruit and small vertebrates…there are several work on this …see for example:

Rosa, C. A., Santos, K. K., Faria, G. M. M., Puertas, F., & Passamani, M. (2015). Dietary behavior of the Maned wolf Chrysocyon brachyurus (Illiger, 1815) and the record of predation of Brown tinamou Crypturellus obsoletus (Temminck, 1815)(Tinamiformes, Tinamidae) at Mantiqueira Mountains, Southeastern Brazil. Boletim da Sociedade Brasileira de Mastozoologia, 72, 7-10).

Aragona, M., & Setz, E. Z. F. (2001). Diet of the maned wolf, Chrysocyon brachyurus (Mammalia: Canidae), during wet and dry seasons at Ibitipoca State Park, Brazil. Journal of Zoology, 254(1), 131-136.


In other words, I. ebu would be similar to Proteles and Chrysocyon because its adaptations for foraging behaviour would be similar to theirs in grasslands and open savannas. Please, note that although the dentition of I. ebu “is seemingly adapted for a more hypercarnivorous lifestyle than other members of Ictitherium.”… in any case, Ithitheres, are not very specialized forms: I. ebu retains a large M1 (and the M2) and a m2 (the m1 talonid is also well developed). As in C. brachyurus, insects, small vertebrates or fruit cannot be discharged in its diet.
Authors can test this simply by using a discriminant analysis with a category where specimens can be classified according foraging behaviour (obviously, Proteles and Chrysocyon have to be together as for example omnivores/insectivores).
By the way…Chrysocyon occurs mainly in grassland-dominated regions but it can be found also in savannah woodlands (as probably I. ebu). Please, see refereces for this within: Queirolo, D., Moreira, J. R., Soler, L., Emmons, L. H., Rodrigues, F. H., Pautasso, A. A., ... & Salvatori, V. (2011). Historical and current range of the Near Threatened maned wolf Chrysocyon brachyurus in South America. Oryx, 45(2), 296-303.

Additional comments

I have sent a pdf copy of the ms. with additional comments

·

Basic reporting

Clear, unambiguous, professional English language was used throughout the article. Introduction and background show context on the specimens of interest in this study and appear to be well-referenced and relevant. The figures are relevant, high quality, well-labelled, and described with clear interpretations for the reader. All raw data is presented.

Experimental design

Research is original and fits within the scope of the journal. Research questions are well-defined (i.e., is the extinct hyaenid Ictitherium ebu a cursorial animal as per previous interpretations of other species of Ictitherium), relevant (i.e., understanding past ecosystems within the Miocene of Africa that is defined by a large species turnover during the Miocene-Pliocene shift), and meaningful. It is clrealy stated how the research fills and identified knowledge gap (i.e., the Ecomorphology of one of the carnivores of Lothagam (Kenya)). Rigorous investigation was performed to a high technical and ethical standard (visitation of numerous international collections and various statistical analyses used on different species and different bones and areas of bone known to be correlated with functional morphology/locomotion). Methods were described with sufficient detail and information to replicate (methods used were clearly stated, how they work, and what statistical packages were used for clear replication).

Validity of the findings

Experiment appears to be novel in the application of past works with these methods was applied to a fossil taxon or taxonomic group that was not applied to prior. Conclusion indicates encouragement of future study of Miocene hyaenids in this locality for a better understanding of the area’s past ecology, and its benefit to literature is clearly stated. All underlying data has been provided, and is robust, statistically sound, and controlled. Any issues with data and their interpretation are transparent to the reader. Conclusions were well stated, linked to original research question, and limited to supporting results.

Additional comments

Overall, a solid article. Generally easy and clear for the reader to understand the research question, why it is being asked, what is done to answer it, and what is being interpreted from the results which appear to be sound. The reviewer understands the novelty and importance of this research question in understanding the ecological breadth of a past ecosystem that experienced a species turnover.

There are minor questions and recommendations (mainly about style) from the reviewer:

1. There appears to be an unexpected line return on Line 57, or else an incomplete paragraph separator. See what appears to be the same issue at lines 174, 247, 288, 321, 335, 338, 389, 419, 429, 472, and 515.

2. Consider removing the word “measurement” from the phrase “measurement protocol” to avoid using a variance of the same word as the verb of the sentence in line 149.

3. It is recommended for a little more explanation as to what a truss analysis is during the section starting at line 179. It is briefly mentioned in the introduction starting at line 97 on what it is meant to do (differentiate grappling versus non-grappling predators) This analysis will be more familiar to readers with an engineering or mechanics background than others.

4. With I. ebu predicted to be an aardwolf during the flexible discriminant analysis of the Truss analysis starting at line 456, could it be that the truss analysis results could be correlated to phylogeny? Line 480 mentions that the truss analysis interpreted family as Hyaenidae na dline 482 states that “…the elbow is more similar to Hyaenidae.” Is it possible or worth it to mention that elbow morphology could be more correlated to phylogeny as opposed to overall morphology which appears to be correlated to functional usage?

5. Uncertain why Excel is referred to as just “Excel” in line 162 and “MS Excel” in line 176.

6. Note that overlap in the center of CVA/LDA or PCA plot is expected when there are more than two groups in the former case or when the structure of the data is complex and “modular” in the latter case (lines 333-335). With the CVA shown here, we know that there is one axis for each pair of groups (n-1). The first axis separates one group from the rest, the second separates the second group, and the remaining groups are separated on higher axes which intersect at the origin (0,0) point on the plot (note this description is a simplification in that any one axis can (and in this case does) spread some groups along a spectrum, but generally speaking one expect the pattern we have just describe). In other words, the apparent overlap at the center of the plot may or may not overlap in the full multidimensional LDA space.

7. In the second paragraph of “Non-metric multidimensional scaling” section under “Truss analysis” (starting at line 358), the first two sentences could be one sentence like the third sentence that starts in the middle of line 359.

8. we are not sure if the second paragraph of the Introduction starting at line 45 needs to be in that position or isolated for 2 sentences. There is an unnecessary citation for RStudio on line 197 (just R version citation will do).

CJ Salcido and P. David Polly
Indiana University

---

## Round 0.2 · accepted · Accept

All of the reviewers and I are happy with the new version of the manuscript and feel it addressed all of the previous concerns. While in production, please review the short list of suggestions put forth by Reviewer 2, and then the manuscript will be considered ready for publication.

Reviewer 1 ·

Basic reporting

no comment

Experimental design

no comment

Validity of the findings

no comment

Additional comments

no comment

·

Basic reporting

I think the approach and the conclusions reached are quite reasonable. No doubt this work is important for the scientific community.
I also think the article is now much easier to understand than in its original version. This is a clear case of sometimes ‘less’ is ‘more’. Although some parts of the manuscript are still too technical for a fluent reading, at least they are much easier to follow.

Experimental design

It is correct.

Validity of the findings

Nothing to add to the basic reporting

Additional comments

I only have a few suggestions:

Line 357: I don't see a large overlap in the centre of the graph... what I think is observed is that the aardwolf overlaps with the canids. The rest of the species are more or less separated except for the wolf and the side-striped jackal.

lines 374-377 and lines 415-422: Please, indicate that this is a discriminant analysis for the species somewhere in these paragraphs. The reader does not have to go back to the material and method to deduce that it is a discriminant analysis between species since family, locomotion and habitat are shown later.

lines 409-414: Instead of simply putting the numbers separated by dashes, one could use the expression: distance between landmarks 2 and 5, for example another similar nomenclature. By the way, checking the data in the Truss.xlsx file and in Fig. 4, the lines connecting the landmarks do not correspond to the analysed distances. The distances really analysed could be put in Fig. 4 eliminating those shown in black colour, or put them in another colour leaving the black lines connecting some of the landmarks.

·

Basic reporting

All ok from our perspective.

Experimental design

Revised paper has addressed our concerns.

Validity of the findings

Revised paper has addressed our concerns.

Additional comments

Revised paper has addressed our concerns.